# Molecular and Clinical Relevance of *ZBTB38* Expression Levels in Prostate Cancer

**DOI:** 10.3390/cancers12051106

**Published:** 2020-04-29

**Authors:** Maud de Dieuleveult, Claire Marchal, Anne Jouinot, Anne Letessier, Benoit Miotto

**Affiliations:** 1Team Epigenetics, DNA replication and Cancer, Université de Paris, Institut Cochin, INSERM, F-75014 Paris, France; 2Neurobiology, Neurodegeneration & Repair Laboratory, National Eye Institute, National Institutes of Health, MSC0610, 6 Center Drive, Bethesda, MD 20892, USA; 3Team Genomics and Signalling of Endocrine Tumours, Université de Paris, Institut Cochin, INSERM, U1016, CNRS, UMR8104, F-75014 Paris, France

**Keywords:** *ZBTB38*, prostate cancer, primary tumours, doxorubicin, genomic instability, *SPOP*, disease recurrence

## Abstract

Prostate cancer is one of the most commonly diagnosed cancers in men. A number of genomic and clinical studies have led to a better understanding of prostate cancer biology. Still, the care of patients as well as the prediction of disease aggressiveness, recurrence and outcome remain challenging. Here, we showed that expression of the gene *ZBTB38* is associated with poor prognosis in localised prostate cancer and could help discriminate aggressive localised prostate tumours from those who can benefit only from observation. Analysis of different prostate cancer cohorts indicates that low expression levels of *ZBTB38* associate with increased levels of chromosomal abnormalities and more aggressive pathological features, including higher rate of biochemical recurrence of the disease. Importantly, gene expression profiling of these tumours, complemented with cellular assays on prostate cancer cell lines, unveiled that tumours with low levels of *ZBTB38* expression might be targeted by doxorubicin, a compound generating reactive oxygen species. Our study shows that *ZBTB38* is involved in prostate cancer pathogenesis and may represent a useful marker to identify high risk and highly rearranged localised prostate cancer susceptible to doxorubicin.

## 1. Introduction

Prostate cancer is the second most common cancer in men and the sixth leading cause of death from cancer in men around the world [1,2,3,4,5]. While some men will live with localised good-prognosis prostate tumours that will slowly progress, others will have tumours that progress to advanced stages and die of highly aggressive and metastatic disease [2,6]. The Gleason score, the levels of prostate-specific antigen (PSA) and clinical stage are the most commonly used parameters to assess the clinical trajectory of prostate cancer [7]. Based on these classifications, patients are stratified as low-, intermediate- and high-risk of developing aggressive prostate cancer and metastasis. Nonetheless, more accurate systems need to be implemented to better predict prostate cancers progression and outcome [8,9]. For instance, while Gleason 7 prostate cancer represents intermediate risk prostate cancer, it is shown that Gleason 7 can be further divided in Gleason 7 (4 + 3) and (3 + 4) tumours; with Gleason 7 (4 + 3) tumours associated with worse clinico-pathological outcome, including higher risk of seminal vesicle invasion and positive lymph node, than Gleason 7 (3 + 4) tumours [10]. In addition, a number of studies have reported that genetic and molecular differences exist between tumours from different ethnic groups, and that these differences may account, in combination with lifestyle and socioeconomic-related factors, for differences in prostate cancer prevalence and mortality rate [9,11].

For a better classification of prostate cancer, several clinical and genomic studies started to address the molecular heterogeneity of prostate cancer tumours by profiling and integrating pathological parameters, genomic information and deregulated pathways in large cohorts of localised (i.e., primary) and metastatic prostate cancer [12,13,14,15,16,17,18,19]. These studies revealed recurrent alterations in tumours, including the loss of the tumour suppressor gene phosphatase and tensin-homolog (*PTEN*), gains of the genes encoding transcription factors androgen receptor (*AR*) and proto-oncogene c-Myc (*MYC*), recurrent mutations of genes encoding the cell cycle regulator retinoblastoma associated protein (*RB*), E3 ubiquitin ligase adaptor Speckle Type BTB/POZ Protein (*SPOP*), tumor suppressor p53 (*TP53*), and fusion of E26 transformation-specific family (*ETS*-family) transcription factor genes with androgen-responsive promoters [12,14,15,20,21,22]. Interestingly, some of these features have been proven useful for the classification and care of prostate cancer patients [12,18,23,24,25]. For instance, high levels of genomic instability and copy number alterations in localised tumours are associated with poor disease outcome [12,18,25]. Mutations of *SPOP* and alterations in *ETS* transcription factor *ERG* are mutually exclusive in localised and metastatic tumours, representing different subtypes of tumours [12,15,26]. On the contrary, chromodomain helicase DNA binding protein 1 (*CHD1*) deletion and *SPOP* mutations frequently co-occur in prostate cancer, and these tumours are highly sensitive to androgen biosynthesis inhibitors [24]. Still, the relationship between most of these recurrent alterations remains unclear and do not adequately predict disease outcome or treatment response [12,13,26]. It is thus still necessary to unveil new molecular markers to improve the classification of the different types of prostate cancers.

Genome-wide association studies (GWAS) have identified a single-nucleotide polymorphism in the gene *ZBTB38* (Zinc finger and BTB-domain containing protein 38), linked to the risk for men to develop prostate cancer [27]. *ZBTB38* encodes a transcription factor containing a N-terminus BTB (for Broad-complex, Tramtrack and Bric-a-brac) domain and several C2H2 zinc finger motifs involved in DNA binding [28]. It was initially discovered for its ability to bind methylated DNA sequences [28,29,30]. It is an unstable protein regulated by ubiquitin mechanisms implicating E3 ligase retinoblastoma binding protein 6 (RBBP6) and deubiquitinase USP9X (also known as “fat facets in mammals”) [31,32]. ZBTB38 regulates gene expression and reports show that altering its abundance affects cell differentiation, cell homeostasis and cell survival, and may be involved in the initiation and/or progression of cancer [31,32,33,34,35,36,37,38]. Integrative genomics identified the *ZBTB38* locus as a constituent of the molecular network that may drive triple-negative breast cancer [39]. Knock-down of *ZBTB38* in a neuroblastoma cell line affects the expression of genes involved in autophagy and p53 signalling [37]. In bladder cancer cells, *ZBTB38* promotes cell migration and invasion, in part, by regulating Wnt/β-catenin signalling [38]. It was also showed that down-regulation of *ZBTB38* potentiates the efficacy of anti-cancer DNA demethylating agent 5-azacytidine while causing heightened levels of reactive oxygen species (ROS) [32,33,40]. Interestingly, association between prostate cancer risk and ROS levels has been observed and it is thought that ROS contribute to the development and progression of the disease [41,42,43]. Besides, increase in ROS are traditionally contributing to tissue and DNA damage, and evidence points to additional roles of increase ROS levels in the regulation of cellular processes associated with aberrant growth and proliferation, cell death, cell-cell and cell-matrix communications [44]. However, prevalence of *ZBTB38* expression in prostate cancer remains to be properly documented. 

Here, we tested the clinical significance of *ZBTB38* expression in prostate cancer, based on the analysis of publicly available and freely re-usable large-scale genomic and clinical datasets. We show that *ZBTB38* expression is decreased in localised prostate cancer, and further in metastatic prostate cancer. The low expression of *ZBTB38* is associated with increased genomic instability in localised tumours, which may favour cancer progression. We also show that *ZBTB38* is an independent marker of prognosis. Gene expression profiling further unravelled a gene signature correlated with *ZBTB38* expression in localised tumours that suggests increased efficacy of a ROS-generating drug, doxorubicin, in tumours with low expression levels of *ZBTB38*. Consistently, we demonstrated that depletion of *ZBTB38* in prostate cancer cell lines caused heightened levels of ROS and higher sensitivity to doxorubicin treatment. Our findings suggest that measuring *ZBTB38* mRNA levels in tumours may help better anticipate chromosomal instability in localised tumours, cancer aggressiveness and doxorubicin efficacy.

## 2. Results

### 2.1. ZBTB38 Expression Is Lower in Prostate Cancer Compared to Non-Cancerous Prostate Tissues

To study *ZBTB38* significance in prostate cancer, we exploited publicly available data from several published prostate cancer study cohorts that have both comprehensive molecular and clinico-pathological data. We selected datasets reporting high-throughput gene expression in prostate cancer (localised and metastatic specimens) using benign/normal prostate tissues as controls, with at least five specimens in each group (benign/normal and cancerous) and with gene expression profiling obtained by microarray. We identified nine microarray datasets reporting the expression levels of *ZBTB38* in cancerous and non-cancerous specimens with these criteria [12,14,45,46,47,48,49,50,51]. 

The differential expression of *ZBTB38* between control and prostate cancer specimens was computed separately for each cohort. We found that in eight of these nine cohorts, the expression of *ZBTB38* is significantly lower in localised prostate cancer compared to normal/benign prostate tissues (Table 1). In the four cohorts reporting gene expression data in metastatic prostate cancer, we observed a significant decrease of *ZBTB38* expression levels in metastatic prostate cancer compared to normal/benign prostate tissues, as well as localised prostate cancer (Table 1 and Figure 1a–c). 

To extend our analysis, we also interrogated the TCGA database reporting gene expression profiling obtained by RNA-sequencing in 52 normal prostate specimens and 498 localised prostate cancer [13]. In this cohort, *ZBTB38* expression is also significantly lower in prostate cancer compared to normal prostate tissues (Figure 1d).

*ZBTB38* mRNA levels is thus consistently lower in prostate cancer compare to non-cancerous prostate tissue. In addition, *ZBTB38* expression is further lower in metastatic samples compare to localised tumours.

### 2.2. Association between ZBTB38 Expression and Clinico-Pathological Features of Prostate Cancer

We investigated the association between *ZBTB38* expression levels and clinico-pathological features in prostate cancer patients. We reasoned that *ZBTB38* expression being significantly lower in metastatic samples, including these patients in our analysis would significantly impact on the correlation with clinico-pathological features. We thus decided to consider only localised prostate cancer and cohorts for which there are at least 60 localised tumours analysed. We divided localised prostate cancer patients in two groups: “*ZBTB38*-low” and “*ZBTB38*-high” tumours using the median expression value of *ZBTB38* in each cohort. Importantly, in the four cohorts meeting these criteria, only one patient, in the TCGA cohort (TCGA-EJ-5525-01), presents a mutation in the coding sequence of *ZBTB38* suggesting that there is no confounding effect of *ZBTB38* mutations on its mRNA expression and its correlation with clinico-pathological parameters.

In Taylor’s cohort (GSE21032) [12] we observed no correlation between *ZBTB38* expression levels and patient’s age (*p* = 0.160), seminal vesicle invasion (*p* = 0.725), lymph node involvement (*p* = 0.324), surgical margin status (*p* = 0.343), extra capsular extension (*p* = 0.267) and clinical grade of the tumour (*p* = 0.994) (Table 2). Interestingly, we observed clear significant correlation between low expression of *ZBTB38* and high PSA levels pre-therapy (*p* = 0.013) or at diagnosis (*p* = 0.03) and high pathological grade (i.e., T3/T4) of the tumours (*p* = 0.019). We also found a tendency between low expression of *ZBTB38* and higher Gleason score (*p* = 0.059) (Table 2). 

In Kunderfranco’s cohort (GSE14206) [50], we observed a tendency between low *ZBTB38* expression levels and high Gleason score (*p* = 0.065) and a significant correlation with patient’s age (*p* = 0.044). We also observed that “*ZBTB38*-low” tumours are significantly associated to higher pathological grade (*p* = 0.023) (Table 2).

In Ross-Adams’ cohort (GSE70770) [48], we observed no correlation between *ZBTB38* expression level and patient’s age (*p* = 0.373), extra capsular extension (*p* = 0.54), Gleason score (*p* = 0.918), clinical grade (*p* = 0.681), pathological grade of the tumour (*p* = 0.194) and PSA levels at diagnosis (*p* = 0.289) (Table 2). The only correlation we observed is with surgical margin status (*p* = 0.0003) (Table 2). 

In the TCGA cohort [13] (reporting gene expression values by RNA-sequencing) we observed no correlation between *ZBTB38* expression levels and patient’s age (*p* = 0.584) and PSA levels at diagnosis (*p* = 0.449). By contrast, we observed a significant association of “*ZBTB38*-low” tumours to positive surgical margin status (*p* = 0.014) and high Gleason score (*p* = 0.046) (Table 2).

Our analyses revealed that low expression levels of *ZBTB38* in tumours associates, in several independent cohorts, with higher PSA levels at diagnosis and pathological grade. In addition, these correlations were confirmed by plotting *ZBTB38* mRNA expression levels across Gleason score, PSA levels and pathological grades (Figure 2). It is thus likely that low expression of *ZBTB38* mRNA in localised tumours might contribute to cancer aggressiveness and/or progression. Consistent with this interpretation, many studies have repeatedly reported that higher Gleason score and PSA levels as well as positive margin status are often associated with aggressive prostate cancer [2,12,23,48].

### 2.3. Low Expression of ZBTB38 Correlates with Poor Outcome

We next investigated the prognostic value of *ZBTB38* expression in prostate cancer patients using biochemical recurrence after primary curative treatment as a readout of disease-free survival. Only three cohorts, included in this study, report disease-free survival data (i.e., Taylor, Ross-Adams and TCGA), and one of them includes ethnicity information (i.e., Taylor). Using these different cohorts, we addressed whether *ZBTB38* expression levels would be correlated with disease-free survival in prostate cancer patients.

Low expression levels of *ZBTB38* (below median value) associates with higher prostate cancer recurrence in the TCGA and Taylor cohorts when all patients are considered in the analysis (Figure 3a). Low expression levels of *ZBTB38* are also associated with higher cancer recurrence in patients with intermediate risk, i.e., Gleason score = 7, in the TCGA cohort (Figure 3a). Kaplan-Meier analysis showed that disease-free survival of patients with lower *ZBTB38* expression was significantly shorter in Taylor and TCGA cohorts, as well as in the TCGA cohort when considering solely intermediate risk (Figure 3b). These correlations were not significant for Ross-Adams cohort, although similar tendencies are observed (Figure 3a,b).

We next investigated whether *ZBTB38* expression would provide differential risk stratification in different ethnicities using the Taylor cohort comprising clinical data for 106 white and 26 black/African American men [12]. Disease-free survival data were similar between black/African American and white men (*p* = 0.6) in this cohort (Figure 3c). We also observed that disease-free survival was significantly shorter for white patients with lower *ZBTB38* expression in tumours (*p* = 0.006), and a similar tendency was observed for black/African American patients (Figure 3c).

Finally, we investigated whether *ZBTB38* expression that provide differential risk stratification in Gleason 7 patients in the TCGA cohort (Figure 3a) would differentially stratified Gleason 3 + 4 and 4 + 3 patients. For this analysis, we considered TCGA patients, included in the 2015 publication as well as new TCGA patients available from the cBioPortal interface [13,52]. We observed that low expression of *ZBTB38* correlates with shorter disease-free survival in patients with Gleason 4 + 3 tumours (*p* < 0.001) while there is no clear correlation in patient with Gleason 3 + 4 tumours (Figure 3d). Our analyses revealed that patients with “*ZBTB38*-low” prostate tumours are more prone to disease relapse.

### 2.4. ZBTB38 Expression Level Is an Independent Prognostic Factor of Disease-Free Survival

Our data show a significant association between low *ZBTB38* expression levels and more aggressive and recurrent prostate tumours. We thus used the C-index to measure the prognostic performance of *ZBTB38* mRNA expression. The C-indexes were 0.76, 0.791 and 0.716 in the Taylor, TCGA and Ross-Adams cohorts respectively (Table 3). Low *ZBTB38* expression is thus a good predictor of disease-recurrence, as Gleason score and PSA levels at diagnosis (Table 3). Multivariate cox analyses were also used to examine association between disease-free survival, *ZBTB38* mRNA expression and other clinico-pathological factors. We demonstrated that *ZBTB38* mRNA expression level predicted disease-free survival independently of Gleason score, PSA levels and positive margin status, in the Taylor’s cohort and the TCGA cohort (Table 3). This suggest that monitoring *ZBTB38* mRNA levels might help in the detection of patients with poor prognosis prostate cancer.

### 2.5. Low Expression of ZBTB38 Associates with Genomic Instability in Localised Prostate Cancer

To explore a possible mechanism linking *ZBTB38* low expression with prostate cancer aggressiveness, we investigated whether *ZBTB38* expression levels were associated with specific molecular features of prostate cancer. This analysis was done in Taylor and TCGA cohorts, which integrate large-scale prostate cancer genome characterization [12,13]. We first investigated the correlation between *ZBTB38* expression levels with genome-wide instability and mutational burden.

In the cohort of Taylor et al. (GSE21034) [12] we noticed that tumours with low expression of *ZBTB38* (using the median expression value of *ZBTB38* in the cohort) present high level of genomic instability define by heightened copy number aberrations (*p* = 0.028) (Table 4). Interestingly, substantial copy number aberrations are associated with a poor clinical outcome in this cohort as described for “*ZBTB38*-low” tumours. 

In the TCGA cohort [13] we also found a significant correlation between low expression of *ZBTB38* and somatic copy-number aberration (*p* < 0.00001) as well as with the fraction of the genome altered (*p* = 0.0004) (Table 4). On the contrary we did not observe any significant correlation between *ZBTB38* expression and mutational burden (*p* = 0.113) (Table 4). These results indicate that low expression of *ZBTB38* associates with higher incidence of chromosomal rearrangements in localised tumours but not with mutational burden, that is anyway quite low in localised prostate cancer [12,13,14,15]. These results strengthen the fact that *ZBTB38* may play a role in the prostate carcinogenesis and highlight its utility as a marker of genomic instability in discriminating patient with more aggressive localised prostate cancer.

We further investigated the relationship between *ZBTB38* expression level and specific recurrent mutations and CNAs. A common alteration in prostate cancer is gene fusion, and specifically fusion between androgen-regulated promoters with *ERG* or other *ETS* transcription factors genes. We thus analysed the association of *ZBTB38* expression levels with these common gene fusions. We did not observe correlation between *ZBTB38* expression and the *ETS* transcription factors gene fusions in the three cohorts of patients (Appendix A).

Using the rich genomic annotation of the TCGA cohort, we next examined association of *ZBTB38* expression levels with frequent alterations found in prostate cancer such as mutation or copy number aberration of *SPOP*, *TP53* and *PTEN*. We did not observe a correlation between *ZBTB38* expression levels and *TP53* and *PTEN* loss or mutation (Appendix A). On the contrary we observe a significant correlation between low expression of *ZBTB38* and (hemi- and homozygous) deletions at the *CHD1* (*p* = 0.002), *SPOPL* (SPOP-like) (*p* = 0.00005) and *RB1* (*p* = 0.0002) loci, gain at the *FANCC* locus (*p* = 0.007) and point mutation of *SPOP* (*p* = 0.001) (Appendix A). Interestingly, deletion of *CHD1*, deletion of *SPOPL* and mutation of *SPOP* tend to co-occur in prostate cancer tumours. More importantly *SPOP* mutations/*CHD1* deletions define a molecular subgroup of prostate tumours which is characterised by moderate to high copy number aberration, elevated levels of DNA methylation (methylation cluster 2), homogeneous gene expression profile (mRNA cluster 1) and is mutually exclusive with *ETS* transcription factor gene fusions. Consistently, low expression of *ZBTB38* is correlated with the same integrative Cluster 1 (*p* < 0.00001), DNA methylation cluster 2 (*p* < 0.00001) and mRNA cluster 1 (*p* < 0.00001) (Appendix A) as *SPOP* mutation molecular subgroup of tumours. Our results indicate a strong association between low *ZBTB38* expression in localised tumours with heightened levels of chromosomal instability as well as mutations of *SPOP*, which reinforce the association of *ZBTB38* with aggressive prostate cancer.

### 2.6. ZBTB38 Expression Is Not a Marker of Chromosomal Instability in Metastatic Prostate Cancer

The comparison of clinical and molecular data between localised and metastatic tumours showed similarities and many differences [13,53]. We thus investigated the relationship between *ZBTB38* expression and genomic parameters in metastatic tumours from three different datasets [17,26,53]. We did not find significant association between *ZBTB38* expression levels and patient’s age, the proportion of the genome altered and the count of somatic mutations per tumour in the different datasets (Appendix A). Our analysis revealed that *ZBTB38* expression correlates with overall genomic instability in localised but not in metastatic tumours. This observation suggests that low expression of *ZBTB38* might contribute to increase chromosomal instability at early stages of the disease.

### 2.7. Gene Expression Profiling Reveals a Gene Signature Distinguishing Tumours Expressing Low and High Levels of ZBTB38, and with Differential Sensitivity to Doxorubicin

To further study the biological alterations and pathways associated with *ZBTB38* expression status, we identified differentially expressed genes (*p* < 0.001) between localised tumours expressing low levels of *ZBTB38* and tumours expressing high levels of *ZBTB38*. To do that, we compared the whole-genome expression profile of the localised tumours expressing low levels of *ZBTB38* (1^st^ quartile) and tumours expressing high levels of *ZBTB38* (4^th^ quartile) in four independent cohorts from different research centres [12,14,48,50]. We intersected the data to identify the set of genes commonly differentially expressed in between low and high expressing groups (Figure 4a). Our analysis identified 216 genes differentially expressed between tumours expressing low and high levels of *ZBTB38* in at least three cohorts; and five genes differentially expressed in the four cohorts. These latter genes are the disintegrin metalloprotease (*ADAM10*), the armadillo repeat containing protein 8 (*ARMC8*), the NF-kB regulator COMM domain containing protein 8 (*COMMD8*), the early endosome antigen 1 (*EEA1*) and protein phosphatase 1 B (*PPM1B*). To validate these findings, we investigated the expression of these five genes in the RNA-sequencing datasets produced by TCGA (Appendix A). We observed that the five genes were positively correlated with *ZBTB38* expression in this cohort, indicating that their expression is down-regulated in “*ZBTB38*-low” tumours and strongly suggesting that they are co-regulated with *ZBTB38* in prostate tumours.

Gene ontology analysis of the 216 differentially expressed genes reveals an enrichment for genes involved in cellular macromolecule catabolic process (FDR *q*-value = 6.73 × 10^−8^), proteosomal protein catabolic process (FDR *q*-value = 1.21 × 10^−5^), organelle assembly (FDR *q*-value = 3.48 × 10^−6^) and mitotic cell cycle (FDR *q*-value = 1.43 × 10^−5^) (Figure 4b). An interrogation of the collection of gene sets in the molecular signatures database (MSigDB) further indicates an enrichment for genes regulated upon doxorubicin-induced cells death (*q*-value = 1.13 × 10^−13^) and TRAIL-mediated resistance (*q*-value = 3.1 × 10^−7^). Both pathways involved reactive oxygen species (ROS) mediated cellular damage as part of their mechanism of action [54,55]. Interestingly, it was shown that *ZBTB38* expression is associated to ROS signalling in cancer cells, suggesting a role of *ZBTB38* in the regulation of ROS levels and doxorubicin sensitivity in prostate cancer [32,33]. 

### 2.8. Knock-Down of ZBTB38 Promotes ROS Accumulation, Cell Migration and Doxorubicin Toxicity in Prostate Cancer Cells

We further explored the role of *ZBTB38* in prostate cancer by transfecting siRNA to knock-down *ZBTB38* expression in prostate cancer cell lines, representative of different cancerous and hormonal status. RT-qPCR analysis indicated a strong down-regulation of *ZBTB38* expression 48h post-transfection of the siRNAs previously proven functional (Figure 5a) [31,33]. 

We first investigated the consequences of *ZBTB38* knock-down on cell survival using the propidium iodide intake assay by fluorescence-activated cell sorting and on cell migration using transwell assays. We observed that depletion of *ZBTB38* has no significant consequences on cell viability at 48 hours (Figure 5b) while it causes an increase in cell migration in transwell assays compared to control cells (Figure 5c and Appendix A). These in vitro results are consistent with the correlation of “*ZBTB38*-low” tumours with aggressive features of prostate cancer such as seminal vesicle invasion and metastasis in prostate cancer patients. 

We then investigated the expression of the five genes found differentially expressed in the 4 cohorts in between low and high *ZBTB38* expressing groups. RT-qPCR analysis show that *ADAM10*, *ARMC8*, *COMMD8, EEA1* and *PPM1B* were expressed at similar levels in prostate cancer cells transfected with *ZBTB38* and control siRNAs suggesting that these genes are not regulated by *ZBTB38*, but rather co-regulated by a common upstream factor and/or pathway (Appendix A).

We next investigated the consequences of *ZBTB38* knock-down on ROS levels using the 2’,7’-dichlorofluorescin diacetate (DCFDA) probe. We observed that depletion of *ZBTB38* causes an increase in DCFDA levels within the cell (Figure 5d). The depletion of *ZBTB38* thus causes heightened levels of ROS within prostate cancer cells, as observed in other cancer cell types [32]. We also assessed the potential role of *ZBTB38* on doxorubicin sensitivity (Figure 5e). As a control of our experimental set up we showed that LnCAP cells were more sensitive to doxorubicin exposure than PC3 and DU145 cells, as previously reported [56]. In all three cell lines, depletion of *ZBTB38* enhances doxorubicin-induced cell death (Figure 5e). *ZBTB38* is thus involved in the regulation of ROS levels in prostate cancer cells, which may favour genomic instability and carcinogenesis, as well as provide avenues for doxorubicin treatments.

## 3. Discussion

### 3.1. ZBTB38 Expression Correlates with Prostate Cancer Progression and Is an Independent Prognostic Marker

In the present study, we exploited 10 publicly available and well-annotated datasets (including gene expression, genomic information and clinico-pathological data) to unravel a function of *ZBTB38* in prostate cancer. We observed that *ZBTB38* is rarely mutated or rearranged in localised and metastatic prostate cancer. However, we observed that *ZBTB38* has significantly lower level of expression in localised prostate cancer compare to benign tissues and the difference is extended in metastatic prostate cancer, whether *ZBTB38* expression was monitored by micro-array or RNA-sequencing technology.

We have also found that “*ZBTB38*-low” tumours are significantly associated with T3-T4 pathological grade, high PSA levels at diagnosis, positive surgical status, and higher Gleason score. However, these results depend of the cohort analysed. The discrepancies of our results may come from the selection of the patients contributing to each cohort or the preparation (and purity) of the samples analysed by array and sequencing [1]. Nevertheless, our results indicate that *ZBTB38* loss of expression correlates with clinico-pathological features predictive of cancer aggressiveness and advanced stage of the disease. We also notice that our observations, and their tendency, are common between black/African American and white men as well as in patients with Gleason score 4 + 3 and 3 + 4. Interestingly, we also consistently observed an association between *ZBTB38* expression and disease-free survival (DFS) with shorter DFS in the “*ZBTB38*-low” class. Despite the fact that this correlation is (again) variable in different cohorts studied, our results show that patients with “*ZBTB38*-low” prostate tumours are more prone to disease relapse in prostate cancer. We further observed that *ZBTB38* mRNA expression is an independent marker of DFS in prostate cancer. Nonetheless, we are aware of the limited number of patients studied in some of these analyses. Future studies, including additional patients, may yield more robust information regarding *ZBTB38* mRNA association with prostate cancer aggressiveness, particularly in different ethnic populations or for patients with intermediate-risk (i.e., Gleason 7) prostate cancer.

Altogether, these results suggest that monitoring *ZBTB38* mRNA expression level may help identify localised prostate tumours with aggressive features, such as genomic instability and invasive properties. They also grant further studies to address the role of *ZBTB38* in prostate carcinogenesis.

### 3.2. ZBTB38 and Its Paralogs ZBTB4 and KAISO/ZBTB33 Are Involved in Prostate Cancer

Genetic analyses have shown that a polymorphism linked with *ZBTB38* mRNA expression is associated with increased risk of prostate cancer [27]. Here, we observed that *ZBTB38* mRNA expression is an independent prognostic marker of disease-free survival in two out of three cohorts analysed. It suggests that monitoring *ZBTB38* SNP and mRNA status might improve the diagnosis and care of prostate cancer patients. The development of immunochemistry assays to monitor ZBTB38 protein levels in prostate specimens might also be valuable, as the protein is tightly regulated by ubiquitination [31,32,40], and its cellular abundance may differ from its mRNA expression.

*ZBTB38* paralogs*, ZBTB4* and *ZBTB33/KAISO,* are also implicated in the progression of prostate cancer [48,57,58,59]. For instance, increased *ZBTB4* expression in tumours is a prognostic factor for increased survival [58]. Shuttling of ZBTB33/KAISO into the nucleus in prostate cancer cells promotes cell invasion and migration; and in patients this phenomenon is correlated with more aggressive cancer parameters, including the ability to metastasise [57]. It would thus be interesting to investigate whether a combination between expression levels of *ZBTB4*, *ZBTB33/KAISO* and *ZBTB38* may help better categorised tumours with different molecular and clinical behaviour.

### 3.3. ZBTB38 Levels Are Correlated with SPOP Alterations and a Transcriptomic Signature Associated with Doxorubicin Sensitivity

A couple of studies have addressed the function of *ZBTB38* in cancer [37,38]. In bladder cancer, *ZBTB38* promotes migration and invasive growth via modulation of the Wnt/β-catenin signalling pathway [38]. In neuroblastoma cell lines, depletion of *ZBTB38* affects the expression of key cancer pathways including tyrosine receptor kinase signalling, cell cycle and checkpoint genes as well as autophagy-related genes [37]. In here, we unveil a strong association between *ZBTB38* and several clinico-pathological and genomic features of prostate cancer, including a strong association with *SPOP* and *SPOPL* alterations. SPOP and SPOPL are MATH-BTB proteins that share an overall 85% protein sequence identity. It is well-known that BTB domains can homo- and hetero-dimerise to provide protein/protein interaction variety [60,61]. In the case of SPOP, the BTB domain interacts with Cullin-RING E3 ubiquitin ligase enzymes to promote protein substrate degradation [62,63]. Is ZBTB38 a partner or substrate of SPOP? This hypothesis remains to be tested but it grants further investigation of *ZBTB38* as a molecular driver of prostate cancer progression, in relation to *SPOP*/*SPOPL* functions.

A comparison of tumours with high and low expression of *ZBTB38* reveals a signature of 216 genes differentially expressed. Several of these genes are important for prostate cancer progression including metalloproteinases *ADAM9* and *ADAM10* [64,65]. A functional annotation of these genes revealed a cellular function in doxorubicin- and TRAIL-response, two cell-death pathways implicating reactive oxygen species (ROS). This observation was reminiscent of previous findings showing that depletion of *ZBTB38* causes heightened levels of ROS in cells and increased toxicity of hydrogen peroxide [32]. This analysis independently unveils a function of *ZBTB38* in the regulation of ROS and the clinical response to chemotherapies generating ROS. Consistently, we demonstrated that depletion of *ZBTB38* in prostate cancer cell lines leads to increase levels of ROS within cells and increase cell death upon doxorubicin treatment. One can thus speculate that low levels of *ZBTB38* in tumours might be associated with higher levels of ROS causing genomic instability that further fuel cancer progression. Monitoring *ZBTB38* expression level in tumours might thus be useful, in combination with other genetic alterations (i.e., *SPOP/SPOPL*), to target metastatic prostate cancer with doxorubicin-based therapies.

### 3.4. ZBTB38 Expression Correlates with Genomic Instability in Prostate Cancer

We observed that lower expression of *ZBTB38* is associated with higher levels of chromosomal instability and *SPOP* alterations. On the contrary, tumours with high levels of *ZBTB38* are associated with recurrent fusions of *ETS* transcription factor genes and a less rearranged genome. These correlations were observed in different cohorts established by different cancer centres. This suggests that irrespective of sample preparation methodologies or bioinformatics pipelines used to define signal-to-noise ratio information, low expression of *ZBTB38* associates with features of genomic instability, including specific CNAs. 

Genomic instability is a prognostic marker in the progression and recurrence of multiple cancers, including prostate cancer [12,18,25]. As whole-genome approaches are not used in clinical practice, *ZBTB38* expression may represent an easier approach to highlight genomic instability levels commonly linked to prognosis. Interestingly, *ZBTB38* expression and genomic instability are correlated only in localised prostate cancer but not in metastatic ones. This data suggests that low *ZBTB38* expression may favour genomic instability at early stages of the tumourigenesis, notably via ROS accumulation. The potential tumour suppressive function of *ZBTB38* will need additional validation in organoid cultures and in animal models.

## 4. Materials and Methods 

### 4.1. Microarray Databases and Selection of Datasets

Microarray datasets were collected from the GEO repository (https://www.ncbi.nlm.nih.gov/ gds), the cBioPortal for Cancer Genomics portal (http://cbioportal.org) and the web-based cancer genomics resource Oncomine (https://www.oncomine.org/) [66,67,68]. We selected all datasets reporting high-throughput gene expression in localised and metastatic prostate tumours using normal (or benign) prostate tissues as controls, with at least five specimens in each group (benign/normal and cancerous). Twenty-six datasets met these requirements but array platforms often lack probes to monitor *ZBTB38* expression (e.g., GSE6099, GSE62872 and GSE6919). We eventually considered for our analysis the following datasets: GSE21032 (ZBTB38 ID: 4299) [12], GSE14206 (ZBTB38 ID: A_23_P212025) [50], GSE55945 (ZBTB38 ID: 15558733_at) [49], GSE3325 (ZBTB38 ID: 15558733_at) [47], GSE3933 (ZBTB38 ID: IMAGE50920) [45], Oncomine dataset VanajaProstate (ZBTB38 ID:22512_at) [51], GSE68545 (ZBTB38 ID:RC_AA465527) [46], GSE35988 (ZBTB38 ID: A_23_P380076) [14] and GSE70770 (ZBTB38 ID: ILMN_2248843) [48]. A full description of the experimental protocols (including the number of samples, the methods to isolate RNA samples and the technology platform used for gene expression analysis) and clinical review of these datasets is available elsewhere [1] and in the respective GSE reports. 

Gene expression values from the Prostate Cancer and PanCancer TCGA cohorts were retrieved through the cBioPortal for Cancer Genomics portal [13,52]. Expression value were determined by RNA sequencing and expressed as mRNA gene expression (quantified using the RSEM package; https://github.com/deweylab/RSEM). Comparison between normal (*N* = 52) and prostate cancer (*N* = 498) specimen was directly retrieved from the Firebrowse website reporting processed TCGA data (http://firebrowse.org).

### 4.2. Clinical and Genomic Data Correlations

Genomic and clinical data analysed are freely available, re-usable and anonymised. The following detailed parameters were extracted: GEO accession number, PubMed identified (PMID), sample type, sample size, age, cancer clinical and pathological stage, disease-free survival and gene expression values by using the GEO2R from the National Center for Biotechnology Information (NCBI), the cBioPortal for Cancer Genomics and online information associated with each dataset. 

We also downloaded copy number aberrations, recurrent *ETS* transcription factor gene fusions, mutational landscape, DNA methylation patterns and previously performed unsupervised clustering analysis when available [12,13,17,26,53]. Somatic mutations are rare in prostate cancer patients. We solely considered the four genes that are mutated in more than 10 specimens in the TCGA cohort: *SPOP*, *TP53*, *KMT2D* and *KMT2C* [13].

Clinical and genomic correlations were performed by ranking patients according to the median value of *ZBTB38* expression in the respective cohorts. Statistical comparisons of the clinico-pathological and genomic characteristics for significance were performed by the chi-square test. Performance of *ZBTB38* expression, PSA level at diagnosis, Gleason score, SMS and patient age for predicting recurrence was assessed using Harrell’s C-index. A C-index of 0.5 indicates that the model has no discriminant ability whereas a value of 1 indicates that the model perfectly discriminates between the two groups. Cox proportional hazards regression was used for the disease-free survival analyses (DFS). DFS was defined as the time from diagnosis until biological or clinical recurrence. Univariate analysis tested the association of *ZBTB38* expression with DFS. To test the independence of its association with DFS, *ZBTB38* expression was then included in multivariate models with Gleason score, PSA level at diagnosis, SMS and patient age.

Kaplan-Meier curves were generated using the online tool: BiostaTGV (https://biostatgv.sentiweb.fr/). Analysis of disease-free survival was performed by comparing patients with low and high expression of *ZBTB38* (according to median values) by Kaplan-Meier analysis. We considered entire cohorts (localised and metastatic tumours), intermediate risk Gleason 7 tumours and ethnicity information. Ethnicity information is available for Taylor and TCGA cohort, nonetheless only nine black/African American patients contribute to the TCGA cohort and we did not consider this cohort further in our analysis. Similarly, disease-free survival records for patients with localised tumours in the Grasso’s cohort are too limited to perform Kaplan-Meier analysis.

All statistical analyses were performed using R-Studio software (https://rstudio.com/) and Bioconductor (https://www.bioconductor.org/). In all analysis, a *p*-value < 0.05 was considered as statically significant.

### 4.3. Gene Expression Analysis of Localised Tumours and Functional Annotation of Differentially Expressed Genes

Gene expression datasets were filtered to compare localised tumours with highest expression levels of *ZBTB38* (4^th^ quartile) and tumours with lowest expression levels of *ZBTB38* (1^st^ quartile). GEO2R was used to identify differentially expressed genes in the different cohorts, using *p* < 0.001 as cut-off. Five studies report gene expression by array for >50 patients with localised tumours (see Figure 1a). We excluded the Lapointe et al. dataset (GSE3933) because of the poor relationship between probe and gene identifiers. We eventually considered Taylor et al. (GSE21032), Grasso et al. (GSE35988), Kunderfranco et al. (GSE14206) and Ross-Adams et al. (GSE70770). The web-based g-profiler portal (https://biit.cs.ut.ee/gprofiler/gost) was used to homogenise gene names and perform subsequent overlaps between the different lists of differentially expressed genes [69]. 

The list of differentially expressed genes was further validated using the TCGA cohorts (i.e., RNA-sequencing data). Functional and biological annotation were performed using the Molecular Signature Database (MSigDB) [70].

### 4.4. Prostate Cancer Cell Lines: Maintenance and Treatments

LnCAP, PC3 and DU145 prostate cancer cells were cultured according to standard procedures in media recommended by manufacturers (ATCC). Transfection of control and *ZBTB38* siRNA duplexes was performed with Lipofectamine2000 from Invitrogen (Thermo Fisher Scientific, Waltham, MA 02451, USA) according to the manufacturer’s instructions. The duplexes are Stealth RNAi siRNA (Invitrogen): si*ZBTB38* #1 UCAAAUGAAUGAGUCUGCACCUGGU (HSS153846), si*ZBTB38* #2 GGACGACUUUCACAGUG ACACGGUA (HSS153847) and si*ZBTB38* #3 GCUCUGCCGGGAACCUCCACAAAUA (HSS153848). Control siRNA (Stealth RNAi siRNA negative control) was purchased from Invitrogen (Thermo Fisher Scientific, Waltham, MA 02451, USA). siRNA-transfected cells were cultured in 6-well plates at 37 °C under 5% CO_2_ for 48 hours before analysis.

### 4.5. Doxorubicin Sensitivity Tests

Doxorubicin (D1515) was purchased from Sigma-Aldrich (Merck KGaA, Darmstadt, Germany) and resuspended in PBS. Cells were treated with 0.1 µg/mL. 

### 4.6. Gene Expression Analysis

mRNA was isolated using the TRIZOL reagent from Invitrogen (Thermo Fisher Scientific, Waltham, MA 02451, USA) according to a standard protocol. Reverse transcription was performed using the Superscript III reverse transcriptase enzyme as recommended by the manufacturer (Invitrogen). Primers used for qPCR analysis are: *ZBTB38* 5’-ACA CTT GCC GAG CAC TCA TAC-3’ and 5’-GAC GAG GGC GAT CTA TAC AAC T-3’; *ADAM10* 5’-ATG GGA GGT CAG TAT GGG AAT C-3’ and 5’-ACT GCT CTT TTG GCA CGC T-3’; *ARMC8* 5’-TGG GAA GTC TTG CTA TGG GTA-3’ and 5’-GTC TGG GGA CAG GTA GTC CTT G-3’; *COMMD8* 5’-TGG CAT TTG TGG TCG AGC TTA-3’ and 5’-ACG TGC ATC CAT TCT TCT GAT T-3’; *EEA1* 5’-AGC AAC TCC TAT AAA CAC AGT GG-3’ and 5’-AGC AAG ATT AGA CTC TCC TCC AT-3’ and *PPM1B* 5’-CTG AAC GAG AAC CAA GTG CG-3’ and 5’-ACG AAC CTC TTG CAC ATT TGA-3’; and for reference: MAPK14 5’-TGC CGA AGA TGA ACT TTG CGA-3’ and 5’-TCA TAG GTC AGG CTT TTC CAC T-3’; GAPDH 5’-GGG GTC ATT GAT GGC AAC AAT A-3’ and 5’-ATG GGG AAG GTG AAG GTC G-3’ and B2M 5’-GGC TAT CCA CGT ACT CCA AA-3’ and 5’-CGG CAG GCA TAC TCA TCT TTT T-3’. All quantitative data are represented as mean ± SD from triplicate experiments. For comparison between different groups, differences were analysed using a two-tailed student *t*-test.

### 4.7. Cell Parameters Analysis by Fluorescent Activated Cell Sorting (FACS)

Propidium iodide (P4170) and 2′,7′-Dichlorofluorescin diacetate (D6883) were purchased from Sigma-Aldrich (Merck KGaA, Darmstadt, Germany). Propidium iodide (PI) is a membrane impermeant dye that is generally excluded from viable cells. 10 µL of PI staining solution (concentration: 10 μg/mL) was added to 1 × 10^6^ harvested cells in 1 mL PBS just prior to analysis by FACS.

2′,7′-Dichlorofluorescin diacetate (DCFDA) is a cell permeant fluorogenic dye that measures reactive oxygen species within the cell. DCFDA (20 μM) was added for 30 minutes at 37 °C before harvesting the siRNA transfected cells. Cell were then directly processed by FACS.

FACS analysis was performed using a BD FACSCalibur^TM^ flow cytometer (BD Biosciences, San Jose, CA 95131, USA) and the data were analysed using BD CellQuest Software. 20,000 events for each sample were analysed. All quantitative data are represented as mean ± SD from triplicate experiments. For comparison between different groups, differences were analysed using a two-tailed student *t*-test.

### 4.8. Cell Migration Assays

For transwell migration assays, harvested cells (5.00e4) were plated onto the upper chamber of a non-coated cell culture insert (polyethylene terephthalate (PET) membrane; 8 μm pores, Falcon) (CORNING, Corning, NY 14831, USA) and the chamber placed in complete medium. After 16 hours, non-migrated cells on the upper side of the membrane were removed with a cotton swab and the cells on the underside of the membrane were stained with crystal violet and numbered. Three random fields on the membrane were analysed and data reported as average number of cells per field. Experiments were performed on 3 independently transfected pools of cells.

## 5. Conclusions

While some men will develop very aggressive prostate cancer tumours, some others will develop slow-growing and indolent cancers. A collection of molecular and clinical features is routinely used for the classification of patients; still better biomarkers or combination of markers are necessary to improve patients’ care. Here, we show that the expression of the gene *ZBTB38* is a new marker of adverse evolution in localised prostate cancer. *ZBTB38* expression levels may help discriminate aggressive-prone localised prostate tumours with a greater likelihood of tumour spread to lymph node and worst outcome, from tumours who can benefit only from observation. Notably, we show that loss of *ZBTB38* expression may constitute a tumour subclass with increased genomic instability. *ZBTB38* mRNA expression, in combination with other genetic alterations (i.e., *SPOP*/*SPOPL*), might thus be useful to identify localised prostate tumours that will exhibit more aggressive behaviour and target metastasis with doxorubicin-based therapies.

## Figures and Tables

**Figure 1 cancers-12-01106-f001:**
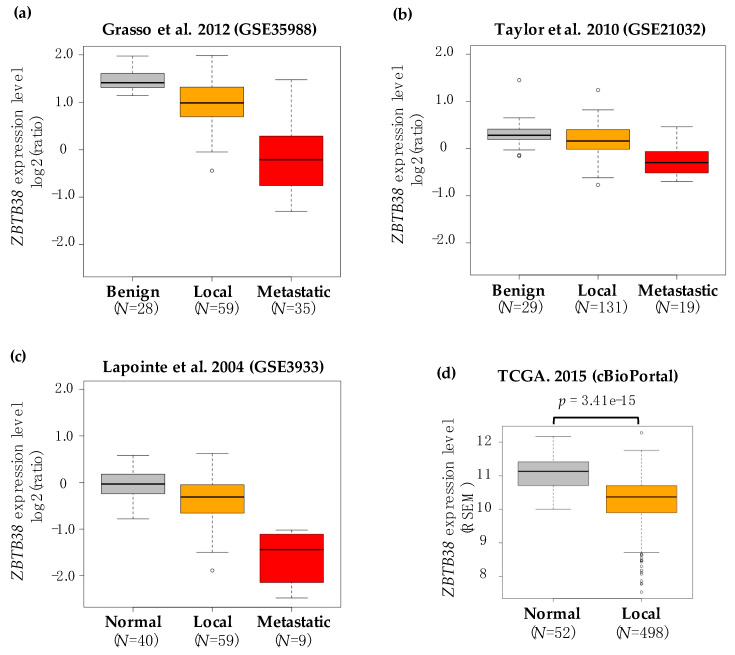
*ZBTB38* expression is lower in prostate cancer compare to non-cancerous tissues. (**a–c**) Box plot showing the levels of *ZBTB38* mRNA expression in benign/normal tissues (in grey), localised tumours (in orange) and metastatic tumours (in red) in three independent prostate cancer cohorts. *ZBTB38* transcript is reported as log2 median-centered ratio. (**d**) Box plot showing the levels of *ZBTB38* mRNA expression in normal tissues (in grey) and localised tumours (in orange) from the TCGA cohorts. *ZBTB38* transcript abundance is quantified using RSEM tool (see Materials and Methods). *N*: number of patients per class.

**Figure 2 cancers-12-01106-f002:**
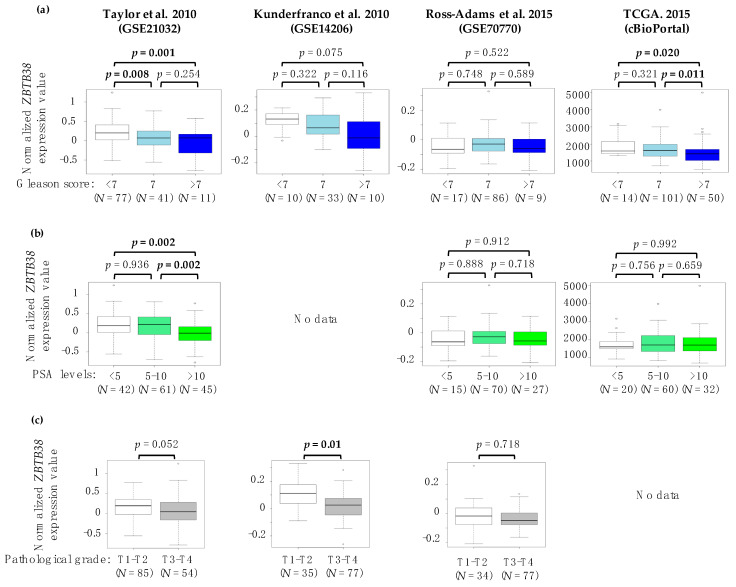
Low expression of *ZBTB38* mRNA is correlated with advanced stage prostate cancer. Correlation between *ZBTB38* mRNA expression and Gleason score (**a**), PSA levels at diagnosis (**b**) and pathological grade (**c**) in prostate cancer. *N*: number of patients per class. Significant *p*-value (*p* < 0.05) are highlighted in bold.

**Figure 3 cancers-12-01106-f003:**
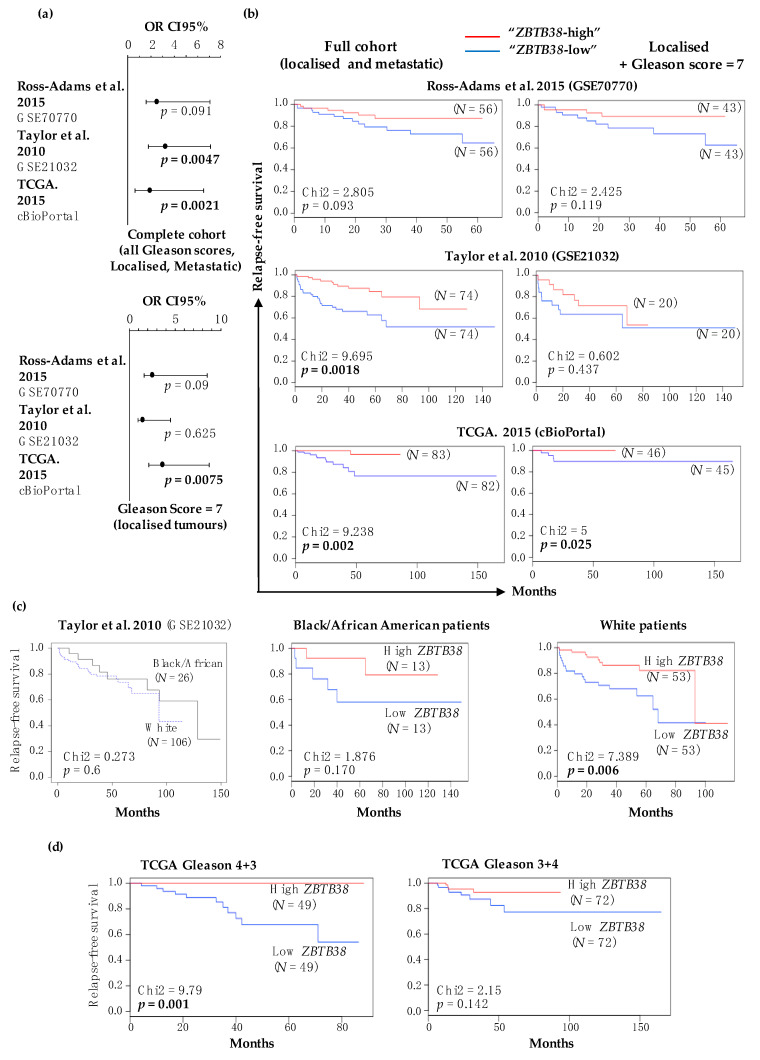
Prognostic significance of *ZBTB38* expression in patients with prostate cancer. (**a**) Association between low levels of *ZBTB38* expression and the relapse. Top panel: study including all patients. Bottom panel: study including patients with intermediate risk cancer (i.e., Gleason score = 7). (**b**) Association between *ZBTB38* expression and disease-free survival in three independent cohorts of patients. Left panel: disease-free survival of all patients according to *ZBTB38* expression levels. Right panel: disease-free survival of intermediate risk (Gleason score = 7) patients according to *ZBTB38* expression levels. (**c**) Association between *ZBTB38* expression and disease-free survival in white and black/African American patients from the Taylor’s cohort. Left panel: disease-free survival of white and black/African American patients. Centre panel: disease-free survival of black/African American patients according to *ZBTB38* expression levels. Right panel: disease-free survival of white patients according to *ZBTB38* expression levels. (**d**) Association between *ZBTB38* expression and disease-free survival of patients with Gleason score 3 + 4 and 4 + 3 from the TCGA cohort. *N*: number of patients per class. Significant *p*-value (*p* < 0.05) are highlighted in bold.

**Figure 4 cancers-12-01106-f004:**
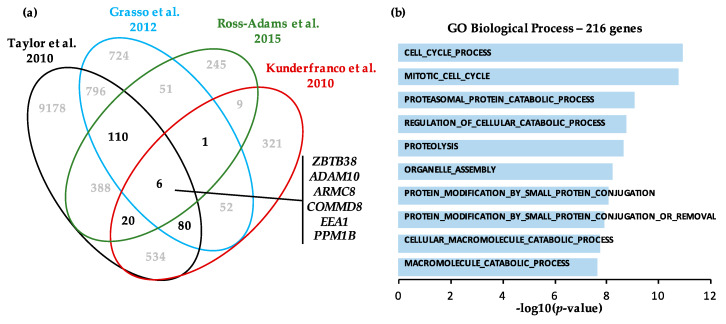
A gene expression analysis unveils a gene signature linked to *ZBTB38* expression and doxorubicin-sensitivity in patients with prostate cancer. (**a**) Venn diagram showing the list of genes differentially expressed in 4 cohorts of patients between tumours expressing low (1st quartile) and high (4th quartile) levels of *ZBTB38*. (**b**) Biological annotation of the gene expression signature, in “*ZBTB38*-high” versus “*ZBTB38*-low” expressing tumours (by using the online MSigDB tool).

**Figure 5 cancers-12-01106-f005:**
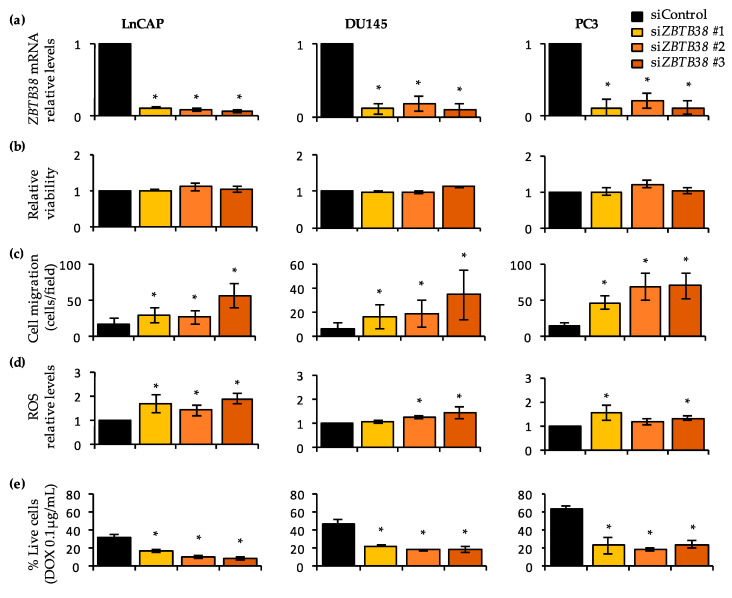
Knock-down of *ZBTB38* promotes ROS accumulation, cell migration and doxorubicin sensitivity in prostate cancer cell lines. (**a**) RT-qPCR validation of *ZBTB38* depletion 48 hours after transfection of control and *ZBTB38* siRNAs in prostate cancer cell lines LnCAP, DU145 and PC3 (*n* = 3). (**b**) Consequence of *ZBTB38* depletion on cell viability monitored by propidium iodide intake, 48 hours after transfection of control and *ZBTB38* siRNAs (*n* = 3). (**c**) Consequence of *ZBTB38* depletion on cell migration assessed by transwell assays, 48 hours after transfection of control and *ZBTB38* siRNAs (*n* = 3). (**d**) Consequence of *ZBTB38* depletion on ROS levels monitored by DCFDA intensity 48 hours after transfection of control and *ZBTB38* siRNAs (*n* = 3). (**e**) Graph reporting the relative proportion of alive prostate cancer cells 48 hours after transfection of control and *ZBTB38* siRNAs and further exposure to doxorubicin (*n* = 3). *, *p* < 0.05, *t*-test (vs. control).

**Table 1 cancers-12-01106-t001:** *ZBTB38* expression in localised and metastatic prostate tumours.

Dataset	Specimen (*N*)			*ZBTB38* Expression				*ZBTB38* ID	Ref.
				Normal/Benign vs. Local		Normal/Benign vs. Metastatic			
	Normal/ Benign	Local	Metastasis	Fold Change	*p*-Value	Fold Change	*p*-Value		
GSE68545	15	15	-	−1.188	**0.039**	-	-	RC_AA465527	[46]
GSE70770	73	113	-	−1.134	**0.01**	-	-	ILMN_2248843	[48]
GSE55945	8	13	-	−1.482	**0.015**	-	-	1558733_at	[49]
GSE14206	14	68	-	−1.062	**0.021**	-	-	21694	[50]
Oncomine	8	27	-	−1.079	0.313	-	-	225512_at	[51]
GSE21032	29	131	19	−1.072	**0.018**	−1.47	**4.52 × 10^−6^**	4299	[12]
GSE35988	28	59	35	−1.261	**3.27 × 10^−4^**	−3.01	**1.31 × 10^−12^**	A_23_P380076	[14]
GSE3933	40	59	9	−1.258	**8.65 × 10^−5^**	−3.28	**2.37 × 10^−17^**	IMAGE:50920	[45]
GSE3325	6	7	6	−1.264	**0.03**	−1.84	**5.1 × 10^−3^**	1558733_at	[47]

*N*, number of patients per class. Significant *p*-value (*p* < 0.05) are highlighted in bold.

**Table 2 cancers-12-01106-t002:** Clinical significance of *ZBTB38* expression in localised prostate tumours.

Characteristics	Taylor et al. 2010 (GSE21032)				Kunderfranco et al. 2010 (GSE14206)				Ross-Adams et al. 2015 (GSE70770)				TCGA. 2015 (cBioPortal)			
	ZBTB38 Expression (N)		Chi2	*p*- Value	ZBTB38 Expression (N)		Chi2	*p*- Value	ZBTB38 Expression (N)		Chi2	*p*- Value	ZBTB38 Expression (N)		Chi2	*p*- Value
	Low	High			Low	High			Low	High			Low	High		
Age (year) at diagnostics																
<50	4	11			3	3			2	4			6	10		
50–60	35	30			22	14			24	17			50	48		
>60	25	24	3.66	0.160	2	9	6.21	**0.044**	30	33	1.96	0.373	62	60	1.07	0.584
Seminal Vesicle Invasion (SVI)																
Negative	56	59			-	-			-	-			-	-		
Positive	7	6	0.12	0.725	-	-	-	-	-	-	-	-	-	-	-	-
Lymph Node Involvement (LNI)																
Normal	46	54			-	-			-	-			-	-		
Abnormal	4	2	0.97	0.324	-	-	-	-	-	-	-	-	-	-	-	-
Surgical Margin Status (SMS)																
Negative	51	48			-	-			35	51			70	87		
Positive	13	17	0.61	0.432	-	-	-	-	21	5	12.8	**0.0003**	43	26	6.02	**0.014**
Extra-Capsular Extension (ECE)																
None	18	23			-	-			16	19			-	-		
Yes	46	41	0.89	0.343	-	-	-	-	40	37	0.37	0.54	-	-	-	-
Gleason Score																
<7	32	45			2	8			8	9			11	8		
7	24	17			18	15			43	43			49	68		
>7	8	3	5.65	0.059	7	3	5.45	0.065	5	4	0.16	0.918	58	42	6.11	**0.046**
Clinical grade																
T1	35	37			-	-			32	29			-	-		
T2	22	24			-	-			18	17			-	-		
T3	2	2	0.01	0.994	-	-	-	-	6	9	0.76	0.681	-	-	-	-
Pathological grade																
T1-T2	38	46			13	22			14	20			-	-		
T3-T4	26	19	5.45	**0.019**	12	5	5.12	**0.023**	42	35	1.68	0.194	-	-	-	-
PSA levels (in ng/mL) at diagnosis																
<5	20	26			-	-			8	5			17	11		
5–10	28	33			-	-			31	39			33	37		
>10	16	5	6.95	**0.03**	-	-	-	-	16	11	2.53	0.289	23	25	1.59	0.449
PSA levels (in ng/mL) pre-treatment																
<5	18	22			-	-			-	-			-	-		
5–10	24	34			-	-			-	-			-	-		
>10	22	8	8.65	**0.013**	-	-	-	-	-	-	-	-	-	-	-	-

*N*, number of patients per class. Significant *p*-value (*p* < 0.05) are highlighted in bold.

**Table 3 cancers-12-01106-t003:** *ZBTB38* expression levels independently predict prostate cancer disease-free survival.

Variable	Taylor et al. 2010 (GSE21032)	TCGA. 2015 (cBioPortal)	Ross-Adams et al. 2015 (GSE70770)
**C-Index for recurrence**			
*ZBTB38* expression	0.76	0.791	0.716
Gleason score	0.842	0.932	0.797
PSA levels	0.680	0.838	0.632
SMS	0.723	0.764	0.628
Patient age	0.519	0.7	0.642
**Multivariate analysis of disease-free survival (HR, [95% CI], *p*-value)**			
*ZBTB38* expression	2.95, [1.44–6.03], **0.003**	4.01, [1.20–13.36], **0.02**	2.19, [0.83–5.79], 0.11
*ZBTB38* + Gleason score	2.01, [0.94–4.01], 0.07	3.82, [1.11–13.15], **0.03**	1.92, [0.72–5.12], 0.19
*ZBTB38* + PSA levels	2.66, [1.28–5.50], **0.008**	2.11, [0.33–13.50], 0.43	2.43, [0.92–6.44], 0.07
*ZBTB38* + SMS	2.93, [1.43–5.98], **0.003**	3.87, [1.15–13.00], **0.03**	2.05, [0.75–5.59], 0.16
*ZBTB38* + Patient age	2.91, [1.42–5.95], **0.003**	3.96, [1.18–13.25], **0.02**	2.19, [0.83–5.77], 0.11

Significant *p*-value (*p* < 0.05) are highlighted in bold.

**Table 4 cancers-12-01106-t004:** *ZBTB38* levels and chromosomal instability in localised prostate tumours.

Characteristics	Taylor et al. 2010 (GSE21032)				TCGA. 2015 (cBioPortal)			
	*ZBTB38* Expression *(N)*		Chi2	*p*-Value	*ZBTB38* Expression *(N)*		Chi2	*p*-Value
	Low	High			Low	High		
Copy Number Aberration Cluster								
Minimal CNA (Class 1–4)	40	43			-	-		
Substantial CNA (Class 5–6)	17	6	4.793	**0.028**	-	-	-	-
SCNA (Somatic Copy number Aberration)								
Quiet	-	-			14	39		
Some SCNA	-	-			62	76		
More SCNA	-	-	-	-	66	31	25.791	**<0.00001**
Fraction Genome Altered								
0	-	-			5	21		
0–0.1	-	-			58	36		
>0.1	-	-	-	-	82	88	15.209	**0.0004**
Mutations								
<20	-	-			18	28		
>20	-	-	-	-	127	118	2.501	0.113

*N*: number of patients per class. Significant *p*-value (*p* < 0.05) are highlighted in bold.

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
