# Peer review of "Molecular and Clinical Relevance of ZBTB38 Expression Levels in Prostate Cancer"

_cancers, 2020, doi:10.3390/cancers12051106_

Round 1

Reviewer 1 Report

Major concerns

I previously recommended that authors must provide human PCa tissue array data, however, they explained that ZBTB38 was an unstable protein in human cancer cells. I cannot agree this interpretation because tissue array sample is fixed by formaldehyde, thus, ZBTB38 is stable in PCa tissue array.

Authors should further provide more solid evidences to support their hypothesis. Analyzing ZBTB38 from online databases just reveal its’ associated relationship in PCa.

Author Response

Reviewer 1

Major concerns

I previously recommended that authors must provide human PCa tissue array data, however, they explained that ZBTB38 was an unstable protein in human cancer cells. I cannot agree this interpretation because tissue array sample is fixed by formaldehyde, thus, ZBTB38 is stable in PCa tissue array.

We fully agree that the stability of ZBTB38 protein as nothing to do with the possibility to perform some tissue array analyses. We actually never claimed that. In our response, we mentioned that ZBTB38 protein is highly unstable and that ZBTB38 protein and mRNA levels do not always correlate. Therefore, the analysis of ZBTB38 protein level in PCa is another question, which will not impact on the correlation between ZBTB38 mRNA levels and tumor aggressiveness, disease recurrence, genomic instability and tumor metastasis, which we report in our present manuscript.

Authors should further provide more solid evidences to support their hypothesis. Analyzing ZBTB38 from online databases just reveal its’ associated relationship in PCa.

We fully agree with the comment that our analysis report an association between ZBTB38 expression level in PCa and different clinical features of PCa. We also show that loss of ZBTB38 leads to the accumulation of ROS (that can induce DNA damage and genomic instability) and to increase cell migration in transwell assays. These in vitro data, even preliminary, show the functional significance of ZBTB38 in ROS and cell migration regulation in PCa and support a potential function of ZBTB38 as oncosuppressor gene in PCa. In addition, our analysis suggests that ZBTB38 levels in tumours might predict doxorubicin susceptibility which we validated in vitro in 3 different PCa cell lines. Thus, our study reports new clinical findings regarding ZBTB38 in PCa and a few functional assays supporting the main correlations using common PCa cell lines.

Reviewer 2 Report

The authors have appropriately answered the questions/concerns raised.

Author Response

The authors have appropriately answered the questions/concerns raised.

Many thanks for your suggestions that clearly improved the clarity and content of the manuscript.

Reviewer 3 Report

The manuscript by de Dieuleveult et al., 'Molecular and Clinical relevance of ZBTB38 Expression levels in Prostate cancer' describes investigations to explore the clinical significance of the zinc finger protein ZBTB38 in prostate cancer. The authors use statistical approach to mine data of published microarray and next generation sequencing data sets of 9 clinical prostate cohorts.

The study as initially submitted however lacked supportive in vitro data validating the functional significance of ZBTB38 as a tumour suppressor.

The authors have worked on this and the manuscript has now has markedly improved with addition of in vitro data using siRNA mediated knockdown of ZBTB38 in cell culture showing the significance of ZBTB38 expression on migration, ROS expression and overall survival of prostate cancer cells.

I would therefor support publication of the study in Cancers.

Author Response

The manuscript by de Dieuleveult et al., 'Molecular and Clinical relevance of ZBTB38 Expression levels in Prostate cancer' describes investigations to explore the clinical significance of the zinc finger protein ZBTB38 in prostate cancer. The authors use statistical approach to mine data of published microarray and next generation sequencing data sets of 9 clinical prostate cohorts.

The study as initially submitted however lacked supportive in vitro data validating the functional significance of ZBTB38 as a tumour suppressor.

The authors have worked on this and the manuscript has now has markedly improved with addition of in vitro data using siRNA mediated knockdown of ZBTB38 in cell culture showing the significance of ZBTB38 expression on migration, ROS expression and overall survival of prostate cancer cells.

I would therefor support publication of the study in Cancers.

We appreciate your support and insightful comments.

Reviewer 4 Report

The manuscript entitled "Molecular and Clinical Relevance of ZBTB38 Expression Levels in Prostate Cancer" provides interesting and translationally significant information of main clinical relevance on the value of zinc transporter ZBTB38 in prostate cancer progression to advanced recurrent disease. The approach is essentially descriptive as the authors perform analysis of available and re-usable large-scale genomic and clinical data sets to determine the levels of ZBTB38 during prostate cancer progression to metastatic disease and define its potential clinical value as an independent marker of tumor progression. The two major weaknesses of the study are the limited conceptual innovation and the lack of a molecular platform/insights into the mechanistic role of the zinc finger in prostate cancer. Previous evidence based on genetic analysis has shown that a polymprphism linked with ZBTB38 mRNA exprression is associated with increased risk of prostate cancer. 

While the study is correlative by nature the authors should exploit the  molecular/genetic events that lead to the functional consequences of loss of the protein during prostate tumor progression to metastasis. They state low expression of ZBTB38 is associated with genomic instability. 

As a minor point the manuscript requires considerable shortening to improve the scientific flow and enable the reader to fully appreciate the results from this analysis. The number of Figures is also large and reduction is required.  

Author Response

The manuscript entitled "Molecular and Clinical Relevance of ZBTB38 Expression Levels in Prostate Cancer" provides interesting and translationally significant information of main clinical relevance on the value of zinc transporter ZBTB38 in prostate cancer progression to advanced recurrent disease. The approach is essentially descriptive as the authors perform analysis of available and re-usable large-scale genomic and clinical data sets to determine the levels of ZBTB38 during prostate cancer progression to metastatic disease and define its potential clinical value as an independent marker of tumor progression. The two major weaknesses of the study are the limited conceptual innovation and the lack of a molecular platform/insights into the mechanistic role of the zinc finger in prostate cancer. Previous evidence based on genetic analysis has shown that a polymprphism linked with ZBTB38 mRNA exprression is associated with increased risk of prostate cancer. 

We thank the referee for the valuable comments and we agree that further functional assays are needed to properly address the function of ZBTB38 in prostate cancer cells and PCa aggressiveness. Still, our study is the first one to address the clinical relevance of ZBTB38 expression in PCa and we validate the functional significance of ZBTB38 as a potential tumour suppressor by using siRNA knock-down in PCa cells.

The genetic analysis reporting in 2011 the link between ZBTB38 SNP and PCa does not address the function of ZBTB38 in PCa. Is the polymorphism in ZBTB38 linked to tumour aggressiveness, genomic instability, metastasis potential and disease recurrence? Is the polymorphism influencing the progression of PCa? Is the polymorphism associated with doxorubicin susceptibility? None of these questions were addressed in the 2011 study. In addition, the correlation between the ZBTB38 SNP and its mRNA level was based on an analysis of adipose tissue. In contrast, our study shows that loss of ZBTB38 in 3 different PCa cell lines: 1) leads to the accumulation of ROS, 2) leads to increase cell migration, 3) causes heightened susceptibility to doxorubycin. Our study shows the functional significance of ZBTB38 in prostate cancer and support a role in prostate cancer tumorigenesis.

Our study also addresses an important clinical issue. The identification of markers helping to stratify Gleason 7 patients is still ongoing, and our study provide clear evidence that in different independent cohorts ZBTB38 mRNA levels in PCa GS=7 discriminate patients most likely to relapse. This information might be very helpful to clinicians and thus important to publish.

For these different reasons, we believe that our study provides new insights, at the clinical and molecular level, regarding the role of ZBTB38 in PCa and that this study is worst publishing and we have been cautious not to oversell the results of our experiments.

Round 2

Reviewer 1 Report

The authors have answered the questions raised, however, it would be beneficial to provide protein expression level of ZBTB38 during prostate cancer progression.

This manuscript is a resubmission of an earlier submission. The following is a list of the peer review reports and author responses from that submission.

Round 1

Reviewer 1 Report

Authors showed that ZBTB38 was associated with poor 14 prognosis in localised prostate cancer using public database. There are several issues to be solved.

Major

1.To show the association of ZBTB38 with increased levels of chromosomal abnormalities and more aggressive features, authors should perform in vitro experiment.

2. Which cells express ZBTB38 in prosatatectomy specimens? Please perform immunohistochemical staining with prostatectomy specimens.

3.In figure 2, authors showed the association of ZBTB38 with the prognosis. Please perform the multivariate analysis adjusting PSA, Gleason score and other factors.

Reviewer 2 Report

The authors of this manuscript identified ZBTB38 in prostate cancer (PCa) as a novel biomarker and claimed that lower expression of ZBTB38 was correlated to high pathological grade, poor outcome, genomics instability and doxorubicin sensitivity. The authors examined and concluded the correlation between ZBTB38 and clinically meaningful outcomes from online public database, however, their conclusions had several conflicts and incomplete.

Major concerns:

1. The authors analyzed several cohorts of tumors by online public dataset (Fig. 1), but they do not analyze the gene level between normal prostate tissues and primary prostate tumors from TCGA, the biggest cohort of localized PCa. Most important of all, the authors should examine the protein expression level of ZBTB38 between normal and PCa, at least, from any commercial available tissue array. As authors mention that sub-cellular localization of ZBTB38 may serve different role on cellular behavior, the signaling intensity of ZBTB38 should be separately evaluated from cytoplasm or nucleus.

2. Why authors concluded that ZBTB38 expression is not a marker in metastatic PCa based on patients’ age, the proportion of the genome rearranged and mutation count, but discard the key point that expression level of ZBTB38 was the lowest in metastatic PCa.

3. The authors suggested that low ZBTB38 correlated with high PSA level by analysis of Taylor prostate dataset (Table 1), but they finally concluded that ZBTB38 did not correlate with AR activity in localized PCa by additional analyzing the correlation ZBTB38 between 20 AR target genes (Table S5). No doubt, AR signaling played important role during PCa progression, however, not all AR downstream genes had significant impact for PCa. What kinds of 20 AR target genes are more important than PSA, authors should provide the list and describe their important background. Of note, there are no reference revealing if ZBTB38 regulates AR signaling or ZBTB38 is a down-stream gene of AR. The conclusion of correlation between ZBTB38 and AR lack solid evidences.

4. The Authors claimed that PCa patients with ZBTB38-low correlated with poor outcome. The rationale for choosing the Gleason score=7 rather than all patients is not explained (Fig. 2).

5. The gene level of ZBTB38 decreased in PCa, why authors knockdown ZBTB38 rather than over-expression it in LnCAP, DU-145 and PC-3. Do authors compare ZBTB38 protein level between non-malignant prostate epithelial cells and PCa cell lines? Moreover, the results of doxorubicin sensitivity assay seemed ZBTB38 increased drug sensitivity for metastatic PCa cell lines rather than localized PCa cells.

Reviewer 3 Report

A well-written and comprehensive study.

A couple of minor changes to the presentation could be considered:

Table 3 shows only negative results – this could be included in the Supplementary information.

The format of some p-values on p5 and 6 e.g. 3,00E-04, should be changed to be consistent with the rest of the document. The p-value on p8 line 217 is missing the decimal point.

Reviewer 4 Report

In the present manuscript, authors have described the clinical significance of ZBTB38 expression in prostate cancer, based on the publicly available datasets.They showed that expression of ZBTB38 is associated with poor prognosis in localised prostate cancer and could help discriminate  aggressive localized prostate tumours from indolent disease and may help towards doxorubicin-based therapies.

Following are the comments/suggestions for authors:

What are the selection criteria for micro-array datasets? How the expression was normalized in each and across different datasets? Define discreetly the exclusion/inclusion criteria for samples considered? Provide box and whiskers plots for distribution of normalized expression level across grade and PSA levels for ZBTB38 expression levels. How to account for variability in findings across different datasets? Specify if diagnostic PSA is PSA at initial biopsy or repeat biopsy? Specify therapy (surgery, HAT?) Can authors show the data by splitting grade 7 in 3+4 and 4+3? Does it signify the findings? Authors may provide comparison of ZBTB38 expression levels with without the combination of clinical variables (age, grade, PSA levels etc) for predicting poor disease outcome (eg BCR) by using ROC analyses. Why authors choose to select different thresholds for ZBTB38 expression for different analyses eg. high and low expression levels for ZBTB38 for clinico-path analyses but low and high quartile ZBTB38 expression levels for BCR analyses. We know that some of the common prostate cancer mutations eg ERG fusion, PTEN loss are race/ethnicity specific, authors may want to include race stratified differences in ZBTB38 expression levels and if it can inform/conclude towards disease outcome specifically in African-American men. Authors may want to include some preliminary data on ZBTB38 expression in prostate tumor specimen, if possible, to strengthen and independently validate the publicly available datasets.

Reviewer 5 Report

The manuscript by de Dieuleveult et al., 'Molecular and Clinical relevance of ZBTB38 Expression levels in Prostate cancer' describes investigations to explore the clinical significance of the zinc finger protein ZBTB38 in prostate cancer. The authors use statistical approach to mine data of published microarray and next generation sequencing data sets of 9 clinical prostate cohorts. The results presented in figure 1 show that ZBTB38 is frequently down-regulated in early prostate cancer stages and its expression is further decreased when the cancer progresses to the advanced metastatic stages. Further correlation of ZBTB38 expression and clinico-pathological features in 3 of the 9 cohorts confirmed an inverse correlation of ZBTB38 mRNA expression, Gleason score, positive margin status and PSA levels. 

It remains unclear, why only data from localised tumours where considered for this, given the much stronger correlation between ZBTB38 expression and tumour development in the advanced stages of the disease, but, the data provided in Figure 1/Table 1 are convincing and the statistical methods used are appropriate. However, validation of the results by orthogonal assays such as immuno-histochemistry would greatly improve the significance of the study. 

The authors next investigated the prognostic value of ZBTB38 expression and correlated mRNA expression levels to relapse free survival. This was performed in only two 2 of the 3 original cohorts (Taylor et al and Ross-Adams et al) whereas Grasso et al was replaced by the TCGA cohort without justification. The authors conclusion that 'ZBTB38-low' prostate tumours are more prone to disease relapse is only confirmed by the TCGA cohort whereas the obvious tendency in Taylor et al. is not significant and the results of the Taylor cohort are even showing the improved disease free survival in patients suffering from ZBTB38-low tumours, thereby not supporting the working hypothesis.

The authors went on to further investigate the correlation of ZBTB38 expression with frequently mutated prostate cancer markers and genomic instability in 2 of the three cohorts and found an association of ZBTB38 down-regulation with markers of genomic instability in particular mutations in SPOP and SPOPL, but wether this is causal or incidental is not further analysed. Equally vague and not validated is the genomic signature of 5 genes commonly down-regulated in ZBTB38-low tumours and associated with reactive oxygen species (ROS) and doxorubicin sensitivity. Although the found association seems rather weak, RNA interference experiments down-regulating ZBTB38 in three independent prostate cancer where successfully used to confirm ZBTB38 down-regulation alone is sufficient to improve resistance against ROS and enhance doxorubicin sensitivity.

Given the published function of ZBTB38 as an DNA binding factor and transcriptional regulator is regretful to see that the authors did not use transcriptome analysis in these cells to investigate the underlying molecular mechanism, to identify potential ZBTB38 target genes or to even validate the association with the 5 genes found in their data mining endeavour. 

Taken together, the proposed study needs further improvement such as orthogonal assays modulating the expression of ZBTB38 in cell culture and xenograft models to validate the without doubt interesting suggestion of ZBTB38 as a potential tumour suppressor in prostate cancer.